# The Synthesis of Hypodiphosphoric Acid and Derivatives with P-P Bond, including Esters and Diphosphine Dioxides: A Review

**DOI:** 10.3390/molecules26237286

**Published:** 2021-11-30

**Authors:** Jacek E. Nycz

**Affiliations:** Faculty of Science and Technology, Institute of Chemistry, University of Silesia in Katowice, ul. Szkolna 9, PL-40007 Katowice, Poland; jacek.nycz@us.edu.pl

**Keywords:** hypodiphosphoric acid, hypodiphosphoric acid ester, diphosphine dioxide, biological activity, metal complex

## Abstract

The synthesis of hypodiphosphoric acid and its related compounds began in 1877, but no summary of the synthetic efforts has been reported. This review includes published papers related to the molecules containing the >P(=O)-P(=O)< fragment, which notably resembles the structure of the >P(=O)-O-P(=O)< moiety, the essential building block of many important molecules found in nature and in the field of medicinal chemistry. This review covers the strategies related to the synthesis of hypodiphosphoric acid (former name hypophosphoric acid), its ester form, and diphosphine dioxides. Finally, some properties and applications of these structures studied during this period are presented.

## 1. Introduction

Many organophosphorus molecules are biologically active and have been widely used as pesticides in agriculture, anticancer drugs and antiviral agents in medicines, and anthelmintics in the veterinary field. Understanding the mechanism of their reaction or inhibition process is critical to shedding light on the design of more active and efficient compounds in terms of medicinal treatment and agricultural production. Organophosphorus compounds with >P(O)-O-(O)P< structure have been extensively studied. Agents with >P(O)-(O)P< moiety bear the structure apparently resemble to >P(O)-O-(O)P< type of compounds. However, their properties are quite different in terms of of resistance to hydrolysis. It is of interest to explore the impact of the structural similarity and difference on their synthetic pathway and potential applications. So far, only a few papers have been published using the >P(O)-(O)P< moieties to replace >P(O)-O-(O)P< fragment in searching for potentially more effective inhibitors in metabolic routes [1,2], which were nonhydrolyzable analogs of compounds that possess >P(O)-O-(O)P< moieties. A few other examples included mimicking nucleotides.

The phosphorus compounds possessing a P–P bond with R_2_P(O)-(O)PR_2_ type structure are called diphosphine dioxides (R = Alkyl, Aryl), hypodiphosphoric acid ((HO)_2_P(O)-(O)P(OH)_2_; former name hypophosphoric acid) or hypodiphosphoric acid esters ((RO)_2_P(O)-(O)P(OR)_2_). There are only a few analog compounds published, i.e., hypodiphosphonates, in which both substituents on each phosphorus atom are different ((RO)RP(O)-(O)PR(OR)). All the mentioned molecules were insufficiently characterized in terms of spectroscopic characteristics and the procedure of synthesis, probably due to missing general synthetic methodologies, accompanying mechanisms, and/or some limitations in spectroscopic characterization. Other difficulties could be lacking proper structure elucidation for symmetric >P(O)-(O)P< type molecules, e.g., the ambiguity on the molecular formula of hypodiphosphoric acid (HO)_2_P(O)-(O)P(OH)_2_. In contrast, the unsymmetrical >P_A_(O)-(O)P_B_< (P_A_ ≠ P_B_) type compounds are much easier to distinguish due to two inequivalent phosphorus atoms, thanks to the characteristic large ^1^*J*_P,P_ coupling visible in ^31^P{^1^H} NMR spectra and ^n^*J*_P,C_ couplings from both phosphorus atoms visible in ^13^C{^1^H} NMR spectra. Very interesting ^n^*J*_P,H_ couplings from both phosphorus atoms cannot be omitted. The NMR spectroscopy can also be used to differentiate diastereoisomers from each other if they are present.

The present review is intended to comprehensively summarize recent advances in the process of P–P bond formation in the >P(O)-(O)P< type compounds (Figure 1), highlighting plausible mechanistic considerations, defining the scope and limitations, and raising interest in these exciting molecules.

## 2. Hypodiphosphoric Acid

Hypodiphosphoric acid (**1**) has been known about since the end of the 19th century. In 1877, Salzer published a synthetic procedure for the production of hypodiphosphoric acid salts (**1′**) (Figure 1) [3,4,5,6,7].

Until now, no mechanism has been established for the formation of hypodiphosphoric acid (**1**). Acid **1** is a white crystalline molecule, stable as a hydrated form at room temperature. In aqueous solutions molecule **1** behaves as a weak tetrabasic acid at 25 °C (p*K*a_1_ = 2.2, p*K*a_2_ = 2.8, p*K*a_3_ = 7.3, p*K*a_4_ = 10.0) [8,9]. The hypodiphosphates (**1′**) undergo protonation to form compound **1** in strongly acidic aqueous solutions. However, acid **1** is unstable under acidic conditions and disproportionated to H_3_PO_4_ and H_3_PO_3_ [8,10]. One explanation of the instability of molecule **1** is due to a slow rearrangement to unsymmetrical isohypodiphosphoric acid (**2**). In the composition of acid **2**, the two phosphorus atoms in different oxidation states are connected via an oxygen atom (Figure 2) [9].

The acid 2 is not stable and disproportionates to H_4_P_2_O_7_ and H_4_P_2_O_5_. An alternative explanation of its instability could be the tautomerism of isohypodiphosphoric (2) with mixed phosphorus anhydride tautomer (2′) (Figure 2). The tautomeric equilibrium between the pentavalent central phosphorus atom in >P=(O)-H type molecules and the trivalent phosphorus in phosphinous acids R_2_P-OH is almost completely shifted to the side of the formation of a strong P=O bond (Figure 2). The only known example of a thermally stable phosphinous acid is bis(trifluoromethyl)phosphinous acid, (CF_3_)_2_P-OH [11], which was synthesized in 1960 by Burg and Griffiths [12]. Recently, Hoge et al. reported the solvent dependent tautomeric equilibrium between bis(pentafluorophenyl)phosphane oxide and the bis(pentafluorophenyl)phosphinous acid. The phosphinous acid is the dominated form in polar aprotic solvents, such as DMSO or DMF, but bis(pentafluorophenyl)phosphane oxide in toluene or CHCl_3_ [13].

In 1971, the structure of the acid **1** was determined by single-crystal X-ray diffraction measurements [14], followed by structures of alkali metals and heavy alkali-metals [15,16,17,18,19,20,21,22,23,24,25,26,27,28]. Structural analysis of molecule **1′** with organic cations has been undertaken using the salts of protonated adenine [29].

Gjikaj and co-workers synthesized the rubidium and cesium salts of **1′**, the properties of which have been characterized by a combination of several techniques: FT-Raman and ^31^P and ^1^H MAS NMR [18]. They obtained the crystals of isotypic in the triclinic space group *P*-1 with one formula unit in the unit cell. The structures were built up by discrete (H_2_P_2_O_6_)^2-^ and (H_4_P_2_O_6_) units in a staggered conformation for the P_2_O_6_ skeleton and the corresponding alkali-metal cations. X-ray structural analysis confirmed the reported structures, which showed O•••HO-P hydrogen bonds between the (H_2_P_2_O_6_)^2−^ and (H_4_P_2_O_6_) groups, consolidating the systems into a three-dimensional network. The O•••HO-P hydrogen bonds range from 2.444 to 2.551 Å for Rb_2_[(H_2_P_2_O_6_)(H_4_P_2_O_6_)] and from 2.441 to 2.551 Å for Cs_2_[(H_2_P_2_O_6_)(H_4_P_2_O_6_)] molecule. The P–P distance is 2.175 Å for Rb and 2.182 Å for Cs salts, the P–O bond lengths range from 1.501 to 1.569 Å, and both distances are consistent with the results of other works [13,16,24,30,31,32]. Emami et al. synthesized organic-inorganic salts of hypodiphosphoric acid tetraalkylammonium cations, which have been characterized by X-ray crystallography, IR spectroscopy, and NMR measurements in the solid state [30]. The ^31^P{^1^H} CPMAS NMR measurements showed that the Na_4_P_2_O_6_•10H_2_O exhibits only one single resonance in 14.9 ppm, and for Rb salt at 12.5 and 13.5 ppm, and for Cs salt at 11.7 and 11.4 ppm. The ^1^H MAS NMR spectra of Rb salt showed three well-resolved resonances signals at 16.1, 13.9, and 12.2 ppm, and for Cs salts at 16.1, 14.3, and 12.6 ppm. Based on the ^31^P{^1^H} CP-HETCOR spectrum of Rb salt, authors assigned the ^31^P resonance at 13.5 ppm to the dihydrogenhypodiphosphate anion and the resonance at 12.5 ppm to the acid **1** moiety.

The acid **1** and its salts have attracted attention due to the discovery of the ferroelectricity properties of diammonium hypodiphosphate by Szklarz and co-workers [33,34]. Szklarz et al. reported the influence of the hydrogen bond for the ferroelectric phase transition shifts by about 20 K towards higher temperatures. They substituted hydrogen with deuterium in the molecule (NH_4_)_2_H_2_P_2_O_6_, leading to compound (ND_4_)_2_D_2_P_2_O_6_ [35].

## 3. The Hypodiphosphoric Acid Esters

The tetralkyl hypodiphosphoric acid esters (**3**) are known molecules and have been described in literature. However, no analog example with two aryloxy substituents on the pentavalent central phosphorus atom has been reported, especially with spectroscopic characterization. A possible explanation of the inaccessibility of the compounds is the impermanence of the P-O bond with the participation of aryloxy substituents [36]. It is important to mention that the related tetraalkyl hypodiphosphite with both trivalent phosphorus atoms is a unique compound, and tetraaryl hypodiphosphite is not reported in the literature. Even nowadays, we still lack efficient and predictable methods of synthesizing hypodiphosphoric acid esters (**3**) and related compounds. The current reported procedures are not generally practical and frequently suffer from drawbacks, such as low yields, tedious procedures, and lack of functionality tolerance (Figure 3).

Generally, there are four routes for the syntheses of compounds **3** that contain the >P(O)-(O)P< moiety (Figure 3). One route relies on converting dialkyl phosphorohalidate (**5**) (mainly chlorides) by alkali metals in heterogeneous (alkali metals) or homogeneous conditions (e.g., potassium naphthalenide, or alkali metals in liquid ammonia) (method A). The reaction of the electrophiles **5** with alkali metals is a complex transformation, and the data presented in the literature are often incompatible. Baudler [37,38] reported that the treatment of dialkyl phosphorochloridate **5** (formerly dialkyl chlorophosphate) with sodium in an inert solvent, such as xylene, toluene, or diethyl ether, at room temperature followed by gentle warming, resulted in a mixture of products consisting, in the main, of hypodiphosphoric acid esters (**3**) together with considerable amounts of anhydrides **10** (formerly pyrophosphoric acid esters). Similarly, the conversion of electron acceptors **5** by alkali metals in heterogeneous conditions often led to a complex reaction mixture [8]. This is not surprising because alkali metals can react with both starting material **5** and products (e.g., molecules **3** and **10**), which are all electron acceptors [6].

Interestingly, method B presented in Figure 3 relies on the conversion of dialkyl phosphonate **7** (formerly dialkylphosphite) with molecule **5**, which produces tetraalkyl hypodiphosphates **3**, tetraalkyl diphosphates (**10**) (formerly tetraalkylpyrophosphates), and mixed anhydrides **12** [39,40]. Comparing the reaction mixtures from both methods A and B, many similarities could be found, especially between the relative rate of product formation and their distribution profile [10]. For both routes, the mixtures formed are relatively complex.

Stec and co-workers reacted 2-chloro-5,5-dimethyl-1,3,2-dioxaphosphinane 2-oxide (**5a**) with diethyl phosphonate (**7a**) in a benzene solution, in the presence of triethylamine (Figure 4) [41]. Unexpectedly, they isolated only symmetric anhydride, namely 2,2′-oxybis(5,5-dimethyl-1,3,2-dioxaphosphinane 2-oxide) (**10a**) with 82% yield (Figure 4). They did not identify the second possible symmetric anhydride tetraethyl diphosphite (**14a**). This fact is significant for the explanation of the reactivity discussed. The mixed anhydride 5,5-dimethyl-2-oxido-1,3,2-dioxaphosphinan-2-yl diethyl phosphite (**12a**) was possibly formed as an *O*-phosphorylated product in the first step. Next, nucleophile **7a**, in the presence of Et_3_N, produces ammonium salts of 2-hydroxy-5,5-dimethyl-1,3,2-dioxaphosphinane 2-oxide (**16a**) as a leaving group, which in turn reacted with the electrophile **5a**, giving symmetric anhydride **10a** (Figure 4).

In another paper, the same group of authors conducted a reaction between electrophile **5b** and nucleophile **7a** in benzene solution in the presence of triethylamine (Figure 5). They isolated as the main constituent molecule **10b**, contaminated with some amounts of anhydrides of phosphorus acids at different oxidation levels **12b** and **14a**. The authors claimed that compound **3a** was not detected.

The authors in both experiments (Figure 4 and Figure 5) excluded the mixed anhydrides **12a** or **12b** as a key intermediate that could explain the presence of anhydrides **10a** or **10b** “oxidized” and **14a** “reduced” with the absence of molecule **3a**. It was found that under the conditions of the described reaction, the mixed anhydride **12b** did not react with either nucleophile **7a** in the presence of triethylamine at room temperature or electrophile **5a**, which excluded its contribution to the reaction as an intermediate product, a precursor to the observed symmetric anhydrides **10** and **14**. The authors claimed that in Figure 5 the reaction went through the deoxygenation of diethyl phosphorochloridate (**5b**) by diethyl phosphonate (**7a**), which became diethyl hydrogen phosphate (**16b**). The acid **16b** could react with the diethyl phosphorochloridate (**5b**), giving the main product tetraethyl diphosphate (**10b**). On the other hand, the diethyl phosphonate (**7a**) could react with deoxygenated diethyl chlorophosphite (**20a**), giving symmetric three coordinated anhydride tetraethyl diphosphite (**14a**). The acid **16b** could react with electrophile **20a**, giving mixed anhydride **12b** (Figure 6). These researchers demonstrated that anhydrides **10b** and **14a** were not the products of rearrangements of the originally arising anhydride **12b** (Figure 6).

The >P-O^−^ anion (the salts of molecules **7** and **8**) is an ambident nucleophile having a hard centre on the oxygen and a soft one on the phosphorus atom. Accordingly, many examples can be indicated where such an anion reacted selectively, either via the phosphorus [42] or the oxygen [43], respectively (Figure 7).

Similar to the aforementioned reactivity, the ambident phosphorus nucleophiles **7** can react with phosphorus electrophiles **5** to produce both *P*-phosphorylated products, i.e., the tetraalkyl hypodiphosphoric acid esters **3**, and *O*-phosphorylated products i.e., mixed anhydrides **12** (Figure 8). The mixed anhydride **12** is a critical intermediate, explaining the formation of anhydrides **10** (Figure 4 and Figure 8). The pattern of the yield distribution depends on the substituent located on both phosphorus atoms (nucleophile and electrophile). The architecture of products **3** and **12**, with the presence of single or double bonds between oxygen and phosphorus atoms, is controlled by steric and/or electronic factors of substituents R (Figure 8). The reaction mixture analysis indicated that target products 3 were always accompanied by the “reduced” molecules 17 and/or 14 and “oxidized” compounds, resulting in anhydrides 10 (Figure 8).

In 1966, Wall’s team reported the results of an interesting reaction between 3-methyl-5-(trichloromethyl)-1,2,4-oxadiazole and diethyl phosphonate (**7a**) in the presence of triethylamine in a diethyl ether environment [44]. The authors assumed that in the first step of the reaction, nucleophile **7a** attacked the halogen of the trichloromethyl group of the 3-methyl-5-(trichloromethyl)-1,2,4-oxadiazole yielding dehalogenated 5-(dichloromethyl)-3-methyl-1,2,4-oxadiazole with 67% yield. The in situ generated electrophile diethyl phosphorochloridate (**5b**) further reacted with nucleophile **7a,** which was already presented in the reaction environment giving tetraethyl hypodiphosphate (**3a**) and tetraethyl diphosphate (**10b**) (Figure 9). In the next experiment, the authors replaced diethyl ether with ethyl alcohol, increasing the synthesis efficiency of dehalogenated 5-(dichloromethyl)-3-methyl-1,2,4-oxadiazole from 67% to 91% yield. The in situ generated electrophile **5b** reacted faster with ethanol than with molecule **7a**. As a result, the nucleophile **7a** in higher concentration reacted more efficiently with 3-methyl-5-(trichloromethyl)-1,2,4-oxadiazole. The products were isolated by vacuum distillation and identified by vapor phase chromatography.

Steinberg conducted a chlorination reaction of diethyl phosphonate (**7a**) with CCl_4_ in the presence of triethylamine (Figure 10) [45]. He found the formation of high-boiling by-products, which, based on the refractive index values, were assigned a composition of 83% tetraethyl diphosphate (**10b**) and 17% tetraethyl hypodiphosphate (**3a**). The product composition was similar to earlier work undertaken by Nylen [46].

In 1932, Arbuzov et al. treated phosphate anion **7a′** with chlorine gas. In the first step of the reaction, a complex mixture of products was formed, which, after treatment with an excess of chlorine, turned into diethyl chlorophosphite (**20a**) (Figure 11) [47].

Zwierzak et al. carried out a reaction of sodium salt of diethyl phosphonate (**7a′**) and diethyl phosphorochloridate (**5b**) [39]. The authors obtained nucleophile **7a′** through treating diethyl phosphonate (**7a**) by metallic sodium. They noted that the use of 25% excess of nucleophilic reagent **7a′** avoided the formation of anhydride **10b**. After the distillation of the reaction mixture, they obtained a forehead which, from a vigorous reaction with water, attributed to the mixed anhydride **12b**, and the main fraction, based on the refractive coefficient considered to be a mixture of molecule **3a** and anhydride **12b** in a ratio of approximately 5:1 (Figure 12).

Zwierzak’s team reported the efficient synthesis of mixed anhydrides **12** based on the reaction of dialkyl phosphonate **7** with dialkyl phosphorochloridate **5** in benzene, in the presence of pyridine (reaction A; Figure 13) [48]. The products were purified by vacuum distillation. The authors mentioned that they were unable to purify mixed anhydrides **12g**. From the present perspective, the likely explanation of this difficulty is the existence of mixed anhydrides **12c**, **12d**, and **12g** as diastereoisomeric mixtures (meso and rac) among the products (reaction A; Figure 13).

Three years later, in a brief communication, the same group of authors reported the synthesis of tetraethyl hypodiphosphate (**3a**) with a yield of 53% [49]. They reacted sodium salts of diethyl phosphonate (**7a′**) (50% excess) with diethyl phosphorochloridate (**5a**) in benzene solution at 0–5 °C (reaction B; Figure 13). The product was isolated by distillation in vacuo and finally characterized by Raman spectroscopy. The characteristic frequency of the P-P bond was observed at 257 cm^−l^. Additionally, the authors realized that the esterification of anhydrous hypodiphosphoric acid (**1**) by diazoalkanes as an alternative procedure was unsuitable because of the poor accessibility of anhydrous acid **1.**

In a subsequent publication, Zwierzak et al. reported the syntheses of cyclic derivatives of tetralkyl hypodiphosphoric acid ester **3** and its sulfur analogs and related compounds. They mentioned that their cyclic molecules were more stable than acyclic analogs [40]. The described procedure was based on previously reported syntheses of acyclic analogs [49]. They reacted 50% excess of sodium salts of cyclic 5,5-dimethyl-1,3,2-dioxaphosphinane 2-oxide (**7b**) with cyclic 2-chloro-5,5-dimethyl-1,3,2-dioxaphosphinane 2-oxide (**5a**) in benzene solution at 5–10 °C, and anhydride **10a** was not isolated (Figure 14) [40]. For equimolar proportions of used reagents **7b′** and **5a**, the yield of product **3b** decreased to 27%, and small amounts (6%) of anhydride **10a** were isolated.

All yields were referred to products isolated by crystallization and identified by chemical properties, IR spectra, and elemental analysis. The authors compared the stability of acyclic tetralkyl hypodiphosphoric acid esters **3** with obtained molecule **3b**, and found that the acyclic molecules were relatively easily hydrolyzed compared to cyclic **3b**, which remained almost intact when refluxed with water in tetrahydrofuran solution for 20 h.

Mehrotra et al. reported the syntheses of unsymmetric hypodiphosphoric acid esters i.e., diisopropyl dipropyl hypodiphosphate (**3c**), which was obtained in the reaction of diisopropyl phosphorochloridate (**5c**) with the diesters of phosphonic acids activated by the 1,3,2-dioxarsolan moiety ((RO)_2_P(O)—As(OCH_2_)_2_) [50]. Unfortunately, the authors did not present either the synthesis procedure or the characteristics of the resulting products.

Zhao’s team used dialkyl phosphonates **5** (Alkyl = Me, Et, Pr^i^) in the presence of CCl_4_ and triethylamine to synthesize *N*-phosphorylated amino acids and dipeptides [51]. In examining this reaction, they carried out a blank test, i.e., the chlorination of phosphorus nucleophiles **7** in the absence of *N*-nucleophile. However, in the reaction mixture, in addition to the expected dialkyl phosphorochloridates **5**, they observed tetraalkyl diphosphate **10** anhydrides. The reaction presented here is similar to the presented above in Figure 10 by Steinberg [45]. The yields of products **5** and **10** depended on substitutes located on phosphorus atoms.

Rachoń et al. demonstrated the reduction of mixed anhydride 1,3-di-*tert*-butyl-1,3-diphenyldiphosphoxane 1-oxide (mixture of meso and rac) **13a,b** to acid anion **16c′** and *tert*-butyl(phenyl)phosphido anion **19a′** [43]. The obtained intermediates were subsequently treated by sulfur and methyl iodide. The final products *tert*-butyl(phenyl)phosphinic acid (**16c**) and methyl *tert*-butyl(phenyl)phosphinodithioate were isolated with good yields (Figure 15). The presented cleavage is unequivocal and shows that the acid anion **16c****′** is a good leaving group.

The beginning of this section results from a reduction of dialkyl phosphorohalidate (**5**) by alkali metals in heterogeneous (alkali metals) or homogeneous conditions (e.g., potassium naphthalenide or alkali metals in liquid ammonia) were presented (method A; Figure 3). The above results showed that mixed anhydrides **12** can be successfully reduced, and the reduction products were acid anion **16c′** and *tert*-butyl(phenyl)phosphido anion **19a′** (Figure 15). Their presence can explain alternatively the mysterious origin of “oxidized” and “reduced” anhydrides **10** and **14**, respectively, during the synthesis of molecules **3**, which will be further explained.

The reactivity of ambident nucleophiles **7** is much less predictable than nucleophiles **19**, having only one center on the phosphorus atom. The >P-O^−^ **7**′ nucleophilicity depends on the type of substituents located on the phosphorus atom. It is assumed that alkoxy, aryloxy substituents directly connected to phosphorus atoms, should increase their nucleophilicity because they are α-nucleophiles. The alpha effect is the interaction of the two lone pairs on adjacent oxygen atoms raises the HOMO of the anions and makes them better and softer nucleophiles.

## 4. The Isomerism between Molecules Possess >P(=O)-P< and >P-O-P< Fragments

To understand the possibilities and predictability of the syntheses of hypodiphosphoric acid **1**, their esters **3**, diphosphine dioxides **4**, and related compounds **4** with P-P bond, it is necessary to know which substituents stabilize the structure of >P(O)-(O)P< type molecules. The best models for such a study are anhydrides **14**, **15,** and their isomers **17** and **18**, respectively.

Anderson‘s team provided evidence that the diethyl phosphonate (**7a**) gave only *O*-phosphorylated products [52]. They developed the synthesis of tetraethyl diphosphite (**14a**) in a clean and fast (15 min) procedure. The obtained anhydride **14a** with trivalent phosphorus atoms is a stable compound that was not transformed into isomeric diethyl (diethoxyphosphaneyl)phosphonate (**17a**) (Figure 16).

Understanding the equilibrium between anhydride **14a** and molecule **17a** (Figure 16) and their origin is the key to predict the efficiency of the synthesis of hypodiphosphoric acid esters **3**, diphosphine dioxides **4**, and related compounds **4** with P-P bond.

Anderson et al. obtained a stable unsymmetrical anhydride with trivalent phosphorus atom, namely 1,3,2-dioxaphospholan-2-yl diethyl phosphite (**14b**) [53]. The authors did not observe any isomerization processes which could produce molecule **17b** or symmetrization transformations leading to symmetrical anhydrides **14a** or **14c** with identical substituents on both phosphorus atoms (Figure 17).

Crofts’ team synthesized a stable anhydride, 2,2′-oxybis(2,3-dihydro-1*H*-isophosphindole) (**15a**) (Figure 18) [54].

Arbuzov et al. obtained nineteen symmetrical and unsymmetrical anhydrides **14** in the reaction between cyclic dialkyl chlorophosphite **20** and cyclic and non-cyclic sodium salts of dialkyl phosphonates **7′** (Figure 19) [55]. The efficiency after the isolation was 20–69% yield, depending on the construction of both reactants.

Foss et al., in 1980, conducted a series of reactions between metal salts of dialkyl phosphonates **7′** and phosphorus electrophiles **20** (Figure 20) [56]. For the first time, they used reagents with dialkylamino substituents on phosphorus atoms and identified molecules **18** as the final products of this reaction (Method A, Figure 20). However, when a nucleophile was used, which was generated in the reaction between molecules **8** and triethylamine, not only were the compounds **18** isolated, but anhydrides **15** were also obtained (Method B, Figure 20).

These researchers concluded that molecules **18** were the primary products, which were subsequently isomerized into the anhydrides **15** and back. It was found that the rise of temperature increased the concentration of anhydrides **15**. These molecules are easy to be distinguished from each other. The molecules **18** possess characteristic large ^1^*J*_PP_ couplings in 125–290 Hz due to inequivalent phosphorus atoms. The symmetric anhydrides **15** (both phosphorus atoms are the same) were identified based on the singlets. In another work, the same team studied reactions in which one substrate had alkoxy residues, and the other had dialkylamino (Figure 21) [57]. Nucleophilic reagents were generated in the reaction with triethylamine. The authors observed a complicated reaction mixture composed of monoxides in which both substituents on each phosphorus atom are different (e.g., **18a**) and the same (e.g., **18b**) molecules **18** and anhydrides **15**. They explained the symmetrization process by the direct reaction of anhydrides **15** with electrophile **20d**.

These researchers also dealt with another interesting experiment, where the trivalent anhydride tetrabutyl diphosphite (**14d**) was reacted with *N,N,N***′***,N***′***,N***′′***,N***′′**-hexaethylphosphanetriamine in a clean procedure, giving dibutyl (bis(diethylamino)phosphaneyl)phosphonate (**18c**) (Figure 22).

In 1984, the same team discussed the possibility of equilibrium between molecules **18** and **15** [58]. At the temperature of −78 °C, compounds **18** were formed, while anhydrides **15** were formed when a reaction was conducted under thermodynamic control. In this case, the nucleophile was generated by reacting molecule **8** with triethylamine at room temperature. Similar to previous research, the authors observed isomerization processes giving a complicated reaction mixture composed of symmetric and unsymmetric molecules **18** and anhydrides **15** [57]. This was confirmed by the fact that molecules **18** spontaneously isomerized into anhydrides **15**, which were more thermodynamically stable [59]. In 1983, the same team in another paper described two reactions of a new type of the trivalent phosphorus electrophile, i.e., mixed anhydride **21a**, with nucleophiles **7d** and **7e**, respectively [60]. The researchers received stable symmetric **14e** and unsymmetric **14f** anhydrides, which did not isomerize into appropriate molecules **17** (Figure 23). It is important to mention that 2-((4-methyl-1,3,2-dioxaphospholan-2-yl)oxy)benzo[*d*][1–3]dioxaphosphole (**14f**) did not symmetrize.

A few years later, Alfonsov et al. reported an interesting reaction between diethyl (trimethylsilyl) phosphite (**22a**) and diethyl phosphorochloridodithioite (**20e**) (Figure 24). The authors chose silylated phosphorus nucleophiles **22**, and isolated stable compounds **18** with good yields in the range of 44–60%. The reported molecules **18** possessed characteristic large ^1^*J*_PP_ couplings, ^1^*J*_PP_ = 189 Hz for R = EtO and ^1^*J*_PP_ = 163 Hz for R = Me_3_SiO [61]. Isomeric anhydrides **15** was not observed.

Nifantev et al. studied the reaction of cyclic phosphorus reagents 4,6-di-*tert*-butylbenzo[*d*][1–3]dioxaphosphole 2-oxide (**7f**) and 4,6-di-*tert*-butyl-2-chlorobenzo[*d*][1–3]dioxaphosphole (**20f**) in the presence of triethylamine, in which only the sTable 2,2′-oxybis(4,6-di-*tert*-butylbenzo[*d*][1–3]dioxaphosphole) (**14g**) was formed (Figure 25) [62].

The presence of halogen in the phosphorus electrophile constitution could influence side reactions, such as X-philic substitution [63], disproportionation, or symmetrization processes. Due to these drawbacks, another type of reaction of trivalent phosphorus electrophile, with 1*H*-imidazole as a leaving group, was studied, in which the 1*H*-imidazole cyclic phosphate’s derivative 1-(5,5-dimethyl-1,3,2-dioxaphosphinan-2-yl)-1*H*-imidazole (**21b**) was selected (Figure 26). The only identified product was a stable anhydride, 2,2′-oxybis(5,5-dimethyl-1,3,2-dioxaphosphinane) (**14h**) [64].

## 5. P-Centered Radicals Dimerization

The study of reactions involving organophosphorus radicals has a long history. However, the chemistry of the organophosphorus radical dimerization in terms of mechanism, variation of substitution groups, and reaction conditions is still not sufficiently illustrated [65,66]. The phosphinoyl and phosphonyl radicals are non-planar and, as a result, had a variable degree of s-character [66]. From a theoretical point of view, the synthesis of molecules **3** could be realized by the radical dimerization (Figure 3; Method C). Romakhin et al. investigated the reactivity of phosphorus radicals generated electrochemically in acetonitrile solution (ACN) [67] and found that the dimerization process could produce two different types of compounds **3** and **12** (Figure 27).

The metal (sodium and lithium) salts of dialkyl phosphonates **7′** could be electrochemically oxidized, in which the one-electron oxidation of metal salts **7′** was allowed to generate phosphorus radicals >P(O)^•^ (Figure 27). The authors used acetonitrile as a solvent for the electrochemical oxidation processes. It was noticed that the free radicals were generated in the presence of nucleophiles **7′**. A complex product mixture was identified, including anhydrides 10, 12, and 14, and molecules 3 in the minority, which have been confirmed by ^31^P {^1^H} NMR. The origin of anhydrides **14** was the result of the reaction between nucleophiles **7′** and anhydrides **12** (Figure 28), which has already been proved by Nycz et al. [68]. The chemical shifts of molecules **3** are given: tetraethyl hypodiphosphate (**3a**) δ = 8 ppm; tetrapropyl hypodiphosphate (**3c**) δ= 10 ppm, tetraisopropyl hypodiphosphate (**3d**) δ = 6 ppm and tetraisobutyl hypodiphosphate (**3e**) δ = 11 ppm. The authors found that lithium cation could stimulate the formation of P-P bonds, resulting in higher yields of compound **3**.

Zhou and co-workers showed the first copper-catalyzed P-P bond-forming reactions [69]. They found that the copper-catalyzed P-P bond coupling reaction was highly influenced by even slightly changing reaction conditions. The changes could efficiently catalyze an aerobic oxidative dehydrogenative coupling of *H*-phosphonates **7** to selectively produce hypodiphosphoric acid esters **3** or anhydride **10** (formerly pyrophosphoric acid esters), in high yields (Figure 29).

It was claimed that the synthesis of hypodiphosphoric acid esters **3** led through P-centered radical dimerization, and efficiently made P-P coupling products: **3a** (90%), **3b** (90%), (**3d**; 93%), tetrabenzyl hypodiphosphate (**3f**; 94%), tetrabutyl hypodiphosphate (**3g**; 92%) and tetradodecyl hypodiphosphate (**3h**; 91%).

Two years later, Nycz et al. presented work demonstrating the redox reactivity of the metal salts of phosphorus nucleophiles **7′** and **8′** (where R = Ph, Bu^t^, OCH_2_CMe_2_CH_2_O, or EtO and M = Li or Na) in the reaction with 7-amino-2-methylquinoline-5,8-dione and *N*-(2-methyl-5,8-dioxo-5,8-dihydroquinolin-7-yl)acetamide [70]. The nucleophiles **7′** and **8′** participated in a single-electron transfer (SET) to both quinoline-5,8-dione derivatives, generating the short-lived phosphorus-centered radicals which exclusively and efficiently dimerized to produce molecules: **3a** (54%), **3b** (58%), 1,1,2,2-tetraphenyldiphosphane 1,2-dioxide (**4a**) (62%), and meso- and rac- di-1,2-*tert*-butyl-1,2-diphenyldiphosphine 1,2-dioxides (**4b,c**) (88%), with high yields (Figure 30). All charged nucleophiles **7′** and **8′** participated in the single-electron transfer to quinoline-5,8-diones, as was postulated by Bunnett [71] and Russell [72]. It is worth mentioning that the in-situ generated P-centered radicals did not undergo radical addition reactions to quinoline-5,8-diones.

## 6. The Diphosphine Dioxides

The hypodiphosphoric acid esters **3** and the diphosphine dioxides **4** are related molecules, and they are differentiated by the types of substituents located on phosphorus atoms. The hypodiphosphoric acid esters **3** possess alkoxy and/or aryloxy substituents, and diphosphine dioxides **4** have alkyl and/or aryl (Figure 1). Other molecules like hypodiphosphonates **4** have mixed substituents, i.e., alkyl or aryl and alkoxy or aryloxy or others. The substitution groups disposed of direct impact on the feasibility and efficiency of the reactions. Even nowadays, we lack efficient methods for the synthesis of these compounds. The presented data are often incompatible and sometimes contain notable mistakes, e.g., ^31^P NMR chemical shifts of both meso- and rac- di-1,2-*tert*-butyl-1,2-diphenyldiphosphine 1,2-dioxides (**4b,c**) as the same value, namely 48.9 ppm [70,73]. However, their chemical shifts are very far from each other (δ_rac._ = 50.7 and δ_meso_ = 39.8 ppm) [68]. It should be noted here that the syntheses procedures of related diphosphines **23** of the R_2_P-PR_2_ type (R = Alkyl, Aryl) have been described much better, and the results reported in the literature outlined the application, scope, and limitations of the existing protocols [74,75,76].

The difficulty and even the efficiency of the synthesis were attributed to the equilibrium between molecules **15** and **18**, where trivalent central phosphorus atom underwent pentavalent. However, the diphosphoxane (CF_3_)_2_P-O-P(CF_3_)_2_ was only one exception, which was almost completely shifted to the side of the phosphinous acids, with both trivalent phosphorus atoms. The diphosphoxane represented a unique compound showing no rearrangement to a phosphane oxide [13]. The diphosphoxane could be obtained in the reaction between (CF_3_)_2_PI with silver carbonate giving the final product an 80% yield [13].

### 6.1. Phosphinito and Phosphito-Mercuries Metal Complexes Decomposition

From a theoretical perspective, phosphinito and/or phosphito-mercuries metal complexes **9** should be easily decomposed through radical dimerization into the hypodiphosphoric acid esters or the diphosphine dioxides and related molecules (Figure 3). In 1980 Eichbichler et al. reported that bis(*O*-n-butyl-P-phenylphosphonito-P)mercury (**9a**) (possibly a diastereoisomeric mixture of meso and rac) decomposed into the hypodiphosphonic acid ester (**4d**) (possibly mixture of meso and rac) and metallic mercury at the temperature below 263 K, which was characterized by mass spectroscopy [77]. However, they reported that bis(di-*tert*-butylphosphinito-P)mercury (**9b**) was stable and was not decomposed to the expected 1,1,2,2-tetra-*tert*-butyldiphosphane 1,2-dioxide (**4e**) (Figure 31) [78].

Similarly, in 2008, Nycz reported the preparation of phosphinito and phosphito-mercuries [79]. He noticed that during the isolation of final bis(*tert*-butyl(phenyl)phosphinito)mercury (**9c,d**), a 1,2-di-*tert*-butyl-1,2-diphenyldiphosphane 1,2-dioxide (**4b,c**) (mixture of meso and rac) has been isolated, with 12% yield (Figure 32). Unfortunately, the origin of molecules **4b,c** in this experiment is not clear. Systematic studies should be expected where pure metal complexes decompose thermally or under irradiation to produce possible P-centered radicals.

### 6.2. Some Synthetic Approaches of Diphosphine Dioxides

The diphosphine dioxides **4** can be obtained through the same synthetic routes as previously described in hypodiphosphoric acid esters **3** (Figure 3). The first method is a reaction between electrophiles **6**, with electron donors such as the alkali metals in heterogeneous (Li, Na, K) or homogenous conditions (potassium naphthalenide, Li, Na, K in NH_3liq_.). The first application of the Wurtz-Fittig type synthesis for the P-P bond formation was the coupling of PI_3_ with mercury to diphosphorus tetraiodide, reported by Besson at the end of the XIX century [80]. Other metals, such as lithium [81], sodium [82,83,84,85,86], potassium [85,86], and magnesium [87,88], in organic solvents have been successfully used for the coupling of the less reactive but more accessible phosphinous halides.

The reaction of compounds **6** with alkali metals is complicated, and the data presented in the literature are often inconsistent with each other. Horner et al. isolated, in a poor yield, 1,1,2,2-tetraphenyldiphosphane 1,2-dioxide (**4a**) from the reaction mixture of diphenylphosphinic chloride (**6a**) and Li/Hg [89]. On the other hand, in the reaction of compound **6a** with sodium in toluene, the formation of sodium diphenylphosphinite (**8b’**) or diphenylphosphido anion (**19b’**) was observed, depending on the amount of sodium used in this reaction [90]. Inamoto and co-workers conducted several experiments where electrophile **6a** was treated by alkali metals and metal salts in THF [91]. Sasaki et al. described the reduction of electrophile **6a** with SmI_2_, at room temperature in THF. They smoothly converted molecule **6a** into the corresponding compound **8b**. As a by-product, molecule **4a** was isolated with 7% yield (Figure 33) [92].

The presented evidence indicates that the cleavage of P-Cl bonds of molecules **6** with electron donors may occur either by one-electron or two-electron pathways. The reductive cleavage of P-C, as well as P-O bonds, may occur. Rachoń’s team observed the treatment of compounds **6** in THF with one equivalent of potassium metal (with or without the catalysts: naphthalene or 4,4′-ditertbutylbiphenyl) as well as potassium naphthalenide resulted in a complex mixture of products [42,93]. From the theoretical point of view, the electrophile **5** and **6** can accept an electron to form an anion radical, which should collapse into a phosphonyl radical and a chlorine anion. The phosphonyl radical, similar to the phosphorus radicals presented in Figure 27, as a very reactive species, may dimerize to form P-P or P-O-P bond and subsequently accept an electron to form a >P-O^-^ anion followed by further fragmentation, according to the substituents on the phosphorus and the reaction conditions. At this point, it is worth adding that the reagents of the electrophilic, nucleophilic, and radical characters can exist in such a reaction mixture. Tordo’s team also demonstrated that diarylphosphonyl radical can be reduced into diarylphosphinite anion by the electron-rich olefin [65].

The second method of synthesizing compounds **4** is the reaction of electrophiles **6** with appropriate nucleophiles **8,** i.e., diarylphosphine oxides or dialkylphosphine oxides. This chemical transformation is similar to the reaction described above for hypodiphosphoric acid esters **3**, but with a stronger base, e.g., NaH, alkyllithium, aryllithium, etc. (Figure 3; method B).

Hunter et al. described the reaction between chlorodiphenylphosphane (**20g**) and the nucleophile **8b** in the presence and absence of base [94]. The electrophile **20g** reacted with nucleophile **8b** in the presence of base, giving 1,1,2,2-tetraphenyldiphosphane 1-oxide (**18d**). Based on the ^31^P NMR spectroscopy, the authors proved that these reagents reacted to each other in the absence of base, leading to the initial formation of the molecule **18d**, which had a characteristic double-doublet with chemical coupling ^1^*J*_PP_ = 224 Hz. The compound **18d** underwent further transformations with electrophile **20g** to form tetraphenyldiphosphine (**23a**) and molecule **6a**. The electrophile **6a** could be further deoxygenated by compound **8b** to electrophile **20g**, and the molecule **8b** was oxygenated to diphenylphosphinic acid (**16d**). The origin of compounds **16d** and **23a** was explained to be the side reaction products between **18d** and **20g** (Figure 34). These results are consistent with the earlier work of Stec et al.

Majewski et al. reported similar observations that diethylphosphine oxide (**8c**) reacted with in situ produced diethylphosphinic chloride (**6b**) in the reaction of molecule **8c** with CCl_4_ [95]. Among final products, diethylphosphinic anhydride (**11a**) was observed, but 1,1,2,2-tetraethyldiphosphane 1,2-dioxide (**4f**) was not identified. These results are consistent with the earlier work of Steinberg (Figure 10) [45].

Quin et al. presented the efficient synthesis of symmetrical tetraaryldiphosphine dioxides **4** [96]. The appropriate chlorodiarylphosphanes **20** in the presence of tertiary amines and exposed to oxygen and water produced the target compounds **4** (Figure 35). However, when the air was excluded, no compounds **4** were observed. This suggests that the direct reaction product was diphosphine monoxides **18**, which gave target compounds **4** after oxidation. The tetraphenyldiphosphine monoxide (**18d**) was detected in the reaction mixture. As a result of this chemical transformation, they isolated with high yields 1,1,2,2-tetraphenyldiphosphane 1,2-dioxide (**4a**; 79%), 1,1,2,2-tetra-p-tolyldiphosphane 1,2-dioxide (**4g**; 54%), 1,1,2,2-tetrabenzyldiphosphane 1,2-dioxide (**4h**; 56%), 1,2-bis(4-chlorophenyl)-1,2-diphenyldiphosphane 1,2-dioxide (**4i**; 63%), 4,4′-(1,2-dioxo-1,2-diphenyl-1λ^5^,2λ^5^-diphosphane-1,2-diyl)dibenzonitrile (**4j**; 56%). However, the authors did not provide the structural information about compounds **4i** and **4j**, which should appear as a mixture of meso and rac diastereoisomers. The reaction of nucleophile **8b** with electrophile **20g** in the presence of triethylamine gave a mixture of molecule **4a** (initially formed **18d**) and diphenylphosphinic diphenylphosphinous anhydride (**13c**) or diphenylphosphinic anhydride (**11b**) (Figure 35).

Quin et al. were the first to recognize the duality in secondary phosphine oxides **8**′s chemical character because only the existence of *O*-phosphorylation could explain the obtained anhydrides **11b** and **13c** [95]. Ambident phosphorus nucleophiles **8′** of the type >P-O^–^ could react through either the *P* or *O* nucleophilic centre with phosphorus electrophiles **6** or **20** to produce both *P*-phosphorylated products **4** or *O*-phosphorylated compounds **11** mixed with anhydrides **13**. The pattern of the yield distribution depends on the substituent located on the phosphorus atom, which was controlled by steric and/or electronic factors.

Many results reported above showed mysterious “oxidized” and “reduced” anhydrides **11** and **13**, respectively, as the products of the syntheses of molecules **4**. To explain this puzzle, Rachoń’s team performed the reaction of mixed anhydrides **13** with the metal salts of nucleophile **8′**, having the phosphorus atom surrounded by identical substituents. They performed reactions between the mixed anhydrides **13a,b,** and 5,5-dimethyl-1,3,2-dioxaphosphinan-2-yl *tert*-butyl(phenyl)phosphinate (**13d**) and the potassium salts of nucleophiles **7b′** and **8b′**, which possessed the identical substituents located on the trivalent phosphorus atom in the constitution of mixed anhydrides **13** (Figure 36) [68]. It is important to mention that Stec et al. found that the mixed anhydride **12b** did not react with either nucleophile **7a** in the presence of triethylamine at room temperature or electrophile **5a**. The metal salts of nucleophiles **7** and **8** are stronger reagents than those generated in the reaction between molecules **7** and **8** and triethylamine.

The reactions between the metal salts of nucleophiles **7** and **8** with mixed anhydrides **13** were very selective. The trivalent phosphorus atoms were always attacked by nucleophiles **7b****′** or **8c****′,** giving *P*- or *O*-phosphorylated products **18e,f**, and **14h**, respectively, and an acid anion **16c’** as a leaving group (Figure 36). Depending on the nature of substituents, the diphosphine monoxides **18e,f**, or the anhydride **14h** were obtained. Hydrolysis, oxygenation, or sulfuration gave the final products in similar molar yields in both cases. The trivalent phosphorus atom of diphosphine monoxides **18e,f** were oxidized or sulfurated to produced molecules **4b,c** or **4k,l** with good yields. In the next step, acid anions **16c****′** can be reacted easily with phosphorus electrophiles **6** to produce phosphorus acid anhydrides **11** as “oxidized” products. These results are compatible and consistent with the earlier work of Zwierzak et al. (Figure 4, Figure 5 and Figure 6) [40].

In the aforementioned reaction between nucleophile **8′** and electrophiles **6**, two nucleophiles coexisted in the reaction mixture: **8′** and **16′**. The >P-O^−^ anion, nucleophile **8′** as an α-nucleophile should react faster with electrophiles **6** than the acid anion **16′**. Thus, the increase of nucleophile **6′** concentration should increase the amount of molecule **4**. These results are in line with the earlier work of Michalski et al. (25% excess of nucleophilic reagent **7a′**) [39] and (50% excess sodium salts of diethyl phosphonate **7a′**) [49]. According to Pearson’s hard and soft acid and base principle, leaving the group could affect the ratio of *P*- vs. *O*-phosphorylated product [97]. Thus, nucleophilic substitution proceeds faster when the nucleophile and the leaving group are either hard or soft. Consequently, obtaining higher yields of compounds **4** requires modifying the softness of the phosphorus atom substituents. This can be achieved by changing the halogen atom bound to the phosphorus in electrophiles **6**, from Cl through Br to I. A reaction of nucleophiles **7** or **8** with iodine seems to be an exciting idea, which guarantees a temporary excess of the nucleophile. Moreover, iodine, as opposite to bromine, did not quantitatively react with compounds **3**. Quite surprisingly, the reaction of nucleophiles **7′** or **8′** and iodine in THT or THF/NH_3liq_. yielded compounds **3** and **4** efficiently, which obtained diphosphine dioxides **4** in 98% and hypophosphoric acid esters **3** in 48% yields in a facile and convenient way. The reaction of iodine and nucleophiles **7′** or **8′** anion generated in situ proceeded efficiently in both THF and liquid ammonia. Moreover, it appeared that the dissolution rate of solid iodine limited the reactivity of iodine in liquid ammonia at −78 °C. In the reaction mixture *tert*-butyl(phenyl)phosphinic iodide **6b** was isolated with 1% yield. The results, especially from liquid ammonia, suggest that the reaction proceeded via radical mechanism rather than pure nucleophilic substitution. The alkoxy group as a phosphorus substituent reduced the efficiency of compounds **3** because the iodide anion could react with alkoxy substituents (see Arbuzov’s reaction).

The synthesis of unsymmetrical compounds **3** or **4**, in which one fragment would come from nucleophilic molecule **7** or **8** and the second from electrophilic **5** or **6**, was much more complicated. Both reagents can undergo X-philic substitution [63], which leads to the exchange of characters of reagents, and in effect, symmetrical and unsymmetrical compounds **4** (Figure 37).

In 1983, Foss et al. carried out a thermal inducted rearrangement of unsymmetrical molecule **18g** [98] and determined the equilibrium between compounds **18** and **15**. Some interesting rearrangement and symmetrization similar to the above-described reactivity were observed (Figure 38). It was suggested that monoxides **18** preferred the alkyl substituents, but anhydrides **15** alkoxyl. 

## 7. The Analogs of Biologically Active Compounds That Possess a P-P bond

The synthesis of molecules **3** depended on the type of substituents on the phosphorus atom, as is shown above. Elaborate procedures helped to obtain these interesting molecules. However, the syntheses of analogs of biologically active compounds that possess a P-P bond are still challenging.

Zhou et al. synthesized molecule **3i** derivatives of 1-(5-(hydroxymethyl)-2,5-dihydrofuran-2-yl)-5-methylpyrimidine-2,4(1*H*,3*H*)-dione in high yield by using copper as catalyst (Figure 39) [69].

The antitumor activity of hypodiphosphoric acid **3** is potentially crucial in cancer treatment because the synthesis of deoxythymidine monophosphate (dTMP) is critical during rapid cell proliferation [99]. In general, most normal mammalian cells grow slowly and require less dTMP, hence the interruption of its synthesis can selectively kill cancer cells.

Dirheimer succeeded to obtained hypodiphosphorylated derivatives of nucleotides from hypodiphosphoric acid salts (**1**). The limited characterization data about the structure of hypodiphosphoric acid (**1**) hindered the clarity of the results, mainly because of the unavailability of ^31^P NMR data [100,101].

Recently, Sepulveda-Boza published a synthetic procedure about the modified nucleotide of cytidine containing three directly connected phosphorus atoms, i.e., the PA-PB-PC system [8]. The absence of coupling constants between phosphorus atoms in the presented data left room for further clarification about these results. Coupling constants should be distinguished between each connected phosphorus atom.

Setondji et al. obtained adenosine 5′-*O*-hypodiphosphate (**3j**) by DCC-assisted condensation of hypodiphosphoric acid (**1**) with ((3a*R*,4*R*,6*R*,6a*R*)-6-(6-amino-9*H*-purin-9-yl)-2,2-dimethyltetrahydrofuro [3,4-*d*][1,3]dioxol-4-yl)methanol (Figure 40) [101].

Recently, Stec and co-workers presented a brilliant method for forming organohypodiphosphates containing a P–P bond under mild conditions [1]. They obtained hypodiphosphoric acid esters **3k**, **3l,** and **3m** by applying the DBU as a base in the reaction of 2-alkoxy-2-thio-1,3,2-oxathiaphospholanes with nucleophiles **7**, *O*,*O*-dialkyl *H*-phosphonates or *H*-thiophosphonates (Figure 41). 

The structures of molecules **3k**, **3l,** and **3m** have been characterized by ^31^P NMR, ^1^H-NMR, and FAB MS techniques. ^31^P-NMR spectra confirmed the presence of a P–P bond in the constitution of **3**. On ^31^P-NMR spectrum of **3k,** due to inequivalent phosphorus atoms, two pairs of doublets with spin-spin coupling constants ^1^*J*_PP_ = 466 Hz are present. In other experiments, cyclic derivatives of uridine (Ura) **3n** and adenosine (Ade) **3o** were synthesized (Figure 42).

The authors reported the oxathiaphospholane method of P–P bond formation in another example of the synthesis of nucleoside hypodiphosphate derivative **3p** (Figure 43).

The authors also carried out interesting preliminary studies on the biological activity of the compounds they obtained [1]. Guranowski et al. investigated how the biological activity of adenosine 5′-hypodiphosphate **3j** modified the molecule to carry a P-P bond (Figure 40) [2]. None of the HIT-proteins cleaved its P-P bond and liberated pA. They concluded that natural proteins do not recognize compounds **3j** with P-P bond because of their non-existing in living systems. Moreover, they added that, in addition to their previous unpublished observations, molecule **3j** was also refractory to other hydrolases (bacterial alkaline phosphatase, lupin apyrase, and snake venom phosphodiesterase) and that compound **3j** did not inhibit these enzymes. Only snake venom 50-nucleotidase slowly hydrolyzed this compound and released adenosine.

## 8. Conclusions

In this review, the syntheses of the molecules **3** and **4** containing >P(=O)-P(=O)< fragment are described. Despite the lack of a well-developed mechanism, the well-known Salzer’s synthetic procedure of hypodiphosphoric acid (**1**) is constantly used. The acid **1** is unstable under acidic conditions and disproportionate to H_3_PO_4_ and H_3_PO_3_. The P–P distance is close to 2.182 Å, and the P–O bond lengths range from 1.501 to 1.569 Å. The ^31^P{^1^H} CPMAS NMR measurements showed that the hypodiphosphates **1′** exhibit a single resonance in the range from 14.9 ppm to 11.4 ppm. The acid **1** and its salts show ferroelectricity properties.

The structure of related molecules, the hypodiphosphoric acid esters **3**, and the diphosphine dioxides **4** are stabilized by alkyl, aryl, and cyclic substituents which are directly connected to the phosphorus atom. The use of an excess of nucleophiles during the synthesis of molecules **3** and **4** is also beneficial. The lithium cation could better stimulate the formation of P-P bonds than other alkali metals, resulting in higher yields of compounds **3** and **4**, which requires modifying the softness of the phosphorus atom substituents. This can be achieved by changing the substituents, including the halogen atom bound to the phosphorus in the constitution of electrophiles **5** and **6**, from Cl through Br to I.

The oxathiaphospholane method of P–P bond formation during the synthesis of biologically active analogs is an interesting idea that allows the syntheses of further compounds with potential biological activities. However, some authors have concluded that natural proteins did not recognize compounds with P-P bonds because of their non-existing living systems.

The P-centered phosphorus radicals could dimerize to produce compounds **3** and **4**, which are considered precursors of >P(=O)^•^ type radicals. The studies of reactions involving organophosphorus radicals have great potential in organophosphorus and medicinal chemistry.

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
