# Peer review of "The Synthesis of Hypodiphosphoric Acid and Derivatives with P-P Bond, including Esters and Diphosphine Dioxides: A Review"

_molecules, 2021, doi:10.3390/molecules26237286_

Round 1

Reviewer 1 Report

Jacek Nycz’s review focusses on the synthesis of P-P compounds deriving from the hypodiphosphoric acid unit. This is a sort of niche chemistry even in the realm of phosphorus chemistry, which however, in the opinion of this reviewer, is well worth to be reported and summarized in a comprehensive review. Thus, if the motivations inspiring this review are shareable, the outcome of this work is largely disappointing and does not allow to support the publication of the review in Molecules in its present form. Although the literature coverage is apparently complete, there are many drawbacks that make this work not recommendable for being published. Rather, an in-depth revision and a large rewrite of the manuscript is certainly necessary before the paper reaches the high standard necessary for being published in Molecules.

Points to be considered are as follows:

1) The English is rather weak and needs to be fixed up throughout the manuscript. Reading and correcting of the manuscript from a native English speaking is certainly a necessary step to improve the general quality and, therefore, the readability of the manuscript.

2) In the abstract the P(O)-P(O) moiety is compared with the formally related P(O)-O-P(O) unit. However, this comparison is not realistic and to exaggerate the correlation between them seems a sort of daring flight. Therefore, I'd suggest to tune down throughout the paper the meaning of this linkage.

3) page 1, line 21: why does the author use the wording ‘coordination phosphorus compounds’ to delineate these compounds? The word ‘coordination’ in chemistry has a precise meaning which does not apply to these P-P derivatives.

4) page 3, line 93: The word ‘Tellurion’ does not exist. Reference 26 describes thallium compounds.

5) page 3, line 104: Why O...HO-N? It should be O...HO-P.

6) page 3, line 114: Add ‘NMR’ between ‘CPMAS’ and ‘measurements’.

7) page 3, lines 114/115: The sentence ‘The Na4P2O6•10H2O exhibits only one single resonance in 14.9 ppm, Rb at 12.5 and 13.5 ppm and Cs salts at 11.7 and 11.4 ppm’ is rather unclear and should be rephrased.

8) page 3, line 129: The sentence ‘They labeled the substance by isotope exchange reaction to the proofed thesis.’ is also unclear and should be rewritten.

9) page 3, line 135: What does the author mean with the word ‘paltry’

10) page 4, line 165: typo, diphosphatesn

11)The schemes that accompany the text are generally not complete and lack of several important information. For example in Scheme 4 the formation of 8a is accompanied by the elimination of NEt3HCl and the reaction of 8a with 5a takes place in the presence of NEt3. These reagents should be clearly indicated in the scheme. Generally, solvents and temperatures as well as other relevant reaction conditions, should always be indicated in the schemes and, more importantly, a common layout should be used for the schemes and strictly maintained for sall the graphical material of the manuscript. Sometime the reaction solvent, the secondary reaction products and the yields are indicated, some other not. Thus a careful revision of all the schemes is necessary.

12) Page 5, line 185: the use of the word ‘ambident’ is unclear and perhaps inappropriate. Please use a different term.

13) Passim, all over the manuscript: the words or abbreviations of Latin origin should be written in italic. Thus, words like ‘in situ’, ‘i.e.’, ‘e.g.’ needs to be italicized throughout the manuscript. 

14) Page 6, line 210. The description of the reaction of the oxadiazole with 5a is not clearly described. Could the author rephrase and better explain the scheme of this reaction? In particular, which the fate of the oxazole is?

15) Page 7, Scheme 9: Product 3a should be produced in 17%, as from the text, not in <1%, as indicated in the scheme. Please make clear this point.

17) Page 7, line 226: What does the author mean with 'in the first stage'?

18) Page 7, line 228: the name diethylphosphorochloridite which is used here is formally correct. However, because the IUPAC nomenclature is not always adopted in the manuscript, I'd suggest to call compound 13a diethyl chlorophosphite.

19) Page 8, line 253: the indicated reaction temperature is 0-5°C while in scheme 13 it is reported that the reaction takes place at 1 - 5°C. Please, be consistent.

20) Page 8, line 266: the reported reaction takes place at 5 – 10 °C while in Scheme 14 the same reaction occurs at 1 – 5 °C. Please, be consistent.

21) Scheme 15: benzene vs. PhH, again consistency is lacking.

22) Page 10, line 309: Which arsenic/phosphorus compounds were used?

23) Page 10, line 313: typo, syntheses.

24) Page 10, line 324: The statement ‘were quantitatively isolated’ is incorrect as the yields given in scheme 18 are 96% and 75%, respectively.

25) Page 10, Scheme 18: Compounds 8c and 8d, as reported in Scheme 18, are different from the identically numbered compounds as from Scheme 12. This lack of consistency does not help the reader to follow the discussion and must be corrected. In addition, in the explanation of Scheme 18 it is written ‘the reduction of mixed anhydride ....’, and accordingly a two-electron reduction is depicted in the Scheme. However, a more detailed explanation is necessary about the used reducing agent and the reaction conditions as well. 

26) Page 11, line 354: The anion '12a' should be indicated as a tBuPhenylphosphido anion (Scheme 18).

27) Page 11, line 340: we read: ‘and their sulfur or nitrogen analogs’, however no nitrogen compound has been mentioned in the text.

28) Page 12, scheme 19: The reaction cannot be run in benzene at 0 °C as the b.p. of this solvent is about 6 °C!

29) Page 12, line 362: molecule 4b should be 11b.

30) Page 13, line 372 and passim: The author uses often the word ‘received’ in inappropriate way. Please rephrase.

31) Page 13, line 392: typo ‘phosphorous’.

32) scheme 27: compound 15 is actually 15a as R is ethyl in this reaction.

33) Scheme 28: compound 5f should be 5e.

34) Page 15, line 443: Compound 9b should be 9i.

35) Schemes 30 and 31: the captions to these schemes call for a ‘proposed mechanism’, but in reality the sketch does not individuate any mechanism. The same point holds also in other parts of this manuscript where a mechanism picture is mentioned,. In contrast, the discussion about the mechanism is generally only of marginal significance. The same also in scheme  37.

36) Scheme 31: Scheme 31 does not reflect the text that is mentioning the formation of 'a complex product mixture'. However, compounds 7 and 3 are not included in the reaction scheme and species 8 is written among the reagent species.

37) Scheme 34: The conditions for the decomposition of 15a, as well as those used to attempt the decomposition of the related species 15b, should be specified in the Scheme.

A comparison of schemes 34 and 35 shows that the compound numbered as 15b is used to individuate two different species.

38) Page 18, line 531: the sentence ‘a small amount’ is vague; the quantity of  3f,g should be made explicited.

39) Page 18, line 544: The term periododiphosphane is a scarcely used and outdated name to identify P2I4. The correct name is hypodiphosphorous tetraiodide or, perhaps even better, diphosphorus tetraiodide.

40) Page 18, line 550: typo, tertraphenyldiphosphine dioxide.

41) Page 18, line 556: ‘reaction of molecule 4c with lithium in THF at 150 °C’. Please double check the reaction temperature) perhaps 50 °C?).

42) Page 19, line 564: The two organic compounds may be hardly considered as suitable catalysts for this reaction. The effective catalyst should be indicated.

43) Page 19, lines 577 and 578: The specifications ‘where R = Aryl’ and ‘where R = Alkyl’ is redundant and may be easily omitted.

44) Page 19, line 580: wrong abbreviations ‘AlkLi, ArylLi’.

45) Page 23, line 698: ‘using copper as catalyst’, is incorrect as the effective catalyst of the reaction is copper(I) chloride. Details about the catalyzed reaction should be provided.

46) The numbering scheme adopted by the authors does not help the readability of the manuscript. The compounds are generally classified by their P-P or P-O-P architectures and then individuated by adding a letter (or sometime an index!?) to the number. This fact is in principle correct but, when the class of compounds is expanded to include all the letters of the alphabet, as it happens for the hypodiphosphoric acid esters (3), the chosen system becomes very heavy and hard to be followed by the reader. This reviewer would then suggest to reconsider the adopted numbering system. Likely, the use of some extended tables listing the compounds sharing a similar structural motif could be helpful for the reader.

47) In general, as many abbreviations have been used in the manuscript, a clear definition of each of them should be entered after the first use of such abbreviation. A table summarizing ALL the used abbreviations is missing but it is highly necessary and must be included in the new version of this manuscript.

48) The conclusion section is too short and unsatisfactory and does not indicate which could have been the real motivation inspiring this work. The two sentences forming this section do not add anything to the significance of this compendium and do not offer any justification for this review. The section is therefore limited to a pair of tautological sentences which stress the same identical concept without adding any contribution to the discussion. As a whole, this review consists of a long collection of reactions but does not offer to the reader any critical view of the studied subject matter.

49) In references 53, 80, 92 the title of the journal and the volume should be italicized. Refererences 80 and 96 are incomplete as they lack of the title of the cited article.

Author Response

A point-by-point response to comments from reviewers and editor

Reviewer #1

  • The English is rather weak and needs to be fixed up throughout the manuscript. Reading and correcting of the manuscript from a native English speaking is certainly a necessary step to improve the general quality and, therefore, the readability of the manuscript.

Response: Some improvements in the writing have been made. I have revised the whole manuscript carefully and tried to avoid any grammar or syntax errors. Besides, I have asked several skilled authors of English language papers to check the English. Thank you so much for your help. I appreciate it

2) In the abstract the P(O)-P(O) moiety is compared with the formally related P(O)-O-P(O) unit. However, this comparison is not realistic and to exaggerate the correlation between them seems a sort of daring flight. Therefore, I'd suggest to tune down throughout the paper the meaning of this linkage.

Response: The suggested correction has been made. Thank you so much for your help. I appreciate it.

3) page 1, line 21: why does the author use the wording ‘coordination phosphorus compounds’ to delineate these compounds? The word ‘coordination’ in chemistry has a precise meaning which does not apply to these P-P derivatives.

Response: The suggested correction has been made. Thank you so much for your help. I appreciate it.

4) page 3, line 93: The word ‘Tellurion’ does not exist. Reference 26 describes thallium compounds.

Response: The suggested correction has been made. Thank you so much for your help. I appreciate it.

5) page 3, line 104: Why O...HO-N? It should be O...HO-P.

Response: The suggested correction has been made. Thank you so much for your help. I appreciate it.

6) page 3, line 114: Add ‘NMR’ between ‘CPMAS’ and ‘measurements’.

Response: The suggested correction has been made. Thank you so much for your help. I appreciate it.

7) page 3, lines 114/115: The sentence ‘The Na4P2O6•10H2O exhibits only one single resonance in 14.9 ppm, Rb at 12.5 and 13.5 ppm and Cs salts at 11.7 and 11.4 ppm’ is rather unclear and should be rephrased.

Response: The suggested correction has been made. Thank you so much for your help. I appreciate it.

8) page 3, line 129: The sentence ‘They labeled the substance by isotope exchange reaction to the proofed thesis.’ is also unclear and should be rewritten.

Response: The suggested correction has been made. This sentence has been deleted. Thank you so much for your help. I appreciate it.

9) page 3, line 135: What does the author mean with the word ‘paltry’

Response: The suggested correction has been made. Thank you so much for your help. I appreciate it.

10) page 4, line 165: typo, diphosphatesn

Response: The suggested correction has been made. Thank you so much for your help. I appreciate it.

11)The schemes that accompany the text are generally not complete and lack of several important information. For example in Scheme 4 the formation of 8a is accompanied by the elimination of NEt3HCl and the reaction of 8a with 5a takes place in the presence of NEt3. These reagents should be clearly indicated in the scheme. Generally, solvents and temperatures as well as other relevant reaction conditions, should always be indicated in the schemes and, more importantly, a common layout should be used for the schemes and strictly maintained for sall the graphical material of the manuscript. Sometime the reaction solvent, the secondary reaction products and the yields are indicated, some other not. Thus a careful revision of all the schemes is necessary.

Response: The suggested correction has been made. Three schemes from the old version have been deleted, and one, Scheme 34, was added. Thank you so much for your help. I appreciate it.

12) Page 5, line 185: the use of the word ‘ambident’ is unclear and perhaps inappropriate. Please use a different term.

Dear Reviewer, the term “ambident” I following the bellow’s examples. Please see the recent papers dealing with this term, I hope you agree with me:

  1. Robert J. Mayer, Martin Breugst, Nathalie Hampel, Armin R. Ofial, Herbert Mayr, Ambident Reactivity of Phenolate Anions Revisited: A Quantitative Approach to Phenolate Reactivities. J. Org. Chem. 2019, 84, 8837−8858.
  2. Yi-Gui Wang, Ericka C. Barnes, SavaÈ™ Kaya, Vinit Sharma, The Reactivity of Ambident Nucleophiles: Marcus Theory or Hard and Soft Acids and Bases Principle? J. Comput. Chem. 2019, 40, 2761–2777.
  3. Smith, M. B. March’s Advanced Organic Chemistry, 7th ed.; Wiley: Hoboken, NJ, 2013; p 449.
  4. UIPAC 2014 Goldbook definition:

Definition of Ambident by Merriam-Webster

13) Passim, all over the manuscript: the words or abbreviations of Latin origin should be written in italic. Thus, words like ‘in situ’, ‘i.e.’, ‘e.g.’ needs to be italicized throughout the manuscript.

Response: The suggested correction has been made. Thank you so much for your help. I appreciate it.

14) Page 6, line 210. The description of the reaction of the oxadiazole with 5a is not clearly described. Could the author rephrase and better explain the scheme of this reaction? In particular, which the fate of the oxazole is?

Response: The suggested correction has been made, I hope. Thank you so much for your help. I appreciate it.

15) Page 7, Scheme 9: Product 3a should be produced in 17%, as from the text, not in <1%, as indicated in the scheme. Please make clear this point.

Response: The suggested correction has been made. Thank you so much for your help. I appreciate it.

17) Page 7, line 226: What does the author mean with 'in the first stage'?

Response: The suggested correction has been made. Thank you so much for your help. I appreciate it.

18) Page 7, line 228: the name diethylphosphorochloridite which is used here is formally correct. However, because the IUPAC nomenclature is not always adopted in the manuscript, I'd suggest to call compound 13a diethyl chlorophosphite.

Response: The suggested correction has been made. Thank you so much for your help. I appreciate it.

19) Page 8, line 253: the indicated reaction temperature is 0-5°C while in scheme 13 it is reported that the reaction takes place at 1 - 5°C. Please, be consistent.

Response: The suggested correction has been made. Thank you so much for your help. I appreciate it.

20) Page 8, line 266: the reported reaction takes place at 5 – 10 °C while in Scheme 14 the same reaction occurs at 1 – 5 °C. Please, be consistent.

Response: The suggested correction has been made. Thank you so much for your help. I appreciate it.

21) Scheme 15: benzene vs. PhH, again consistency is lacking.

Response: The suggested correction has been made. Thank you so much for your help. I appreciate it.

22) Page 10, line 309: Which arsenic/phosphorus compounds were used?

Response: The suggested correction has been made. Thank you so much for your help. I appreciate it.

23) Page 10, line 313: typo, syntheses.

Response: The suggested correction has been made. Thank you so much for your help. I appreciate it.

24) Page 10, line 324: The statement ‘were quantitatively isolated’ is incorrect as the yields given in scheme 18 are 96% and 75%, respectively.

Response: The suggested correction has been made. Thank you so much for your help. I appreciate it.

25) Page 10, Scheme 18: Compounds 8c and 8d, as reported in Scheme 18, are different from the identically numbered compounds as from Scheme 12. This lack of consistency does not help the reader to follow the discussion and must be corrected. In addition, in the explanation of Scheme 18 it is written ‘the reduction of mixed anhydride ....’, and accordingly a two-electron reduction is depicted in the Scheme. However, a more detailed explanation is necessary about the used reducing agent and the reaction conditions as well. 

Response: The suggested correction has been made. Thank you so much for your help. I appreciate it.

26) Page 11, line 354: The anion '12a' should be indicated as a tBuPhenylphosphido anion (Scheme 18).

Response: The suggested correction has been made. Thank you so much for your help. I appreciate it.

27) Page 11, line 340: we read: ‘and their sulfur or nitrogen analogs’, however no nitrogen compound has been mentioned in the text.

Response: The suggested correction has been made. Thank you so much for your help. I appreciate it.

28) Page 12, scheme 19: The reaction cannot be run in benzene at 0 °C as the b.p. of this solvent is about 6 °C!

Dear Reviewer, the reactions run in benzene solution at 0 - 5 °C [1], or at 5 - 10 °C [2].

  1. Michalski, J.; Zwierzak, A. A Novel Synthesis of Tetra-alkyl Hypophosphates, Proc. Chem. Soc. 1964, 80.
  2. Stec, W.; Zwierzak, A. Cyclic organophosphorus compounds. II. Some sterically hindered cyclic hypophosphate systems and 855 related compounds. Can. J. Chem. 1967, 45, 2513–2520.

29) Page 12, line 362: molecule 4b should be 11b.

Response: The suggested correction has been made. Thank you so much for your help. I appreciate it.

30) Page 13, line 372 and passim: The author uses often the word ‘received’ in inappropriate way. Please rephrase.

Response: The suggested correction has been made. Thank you so much for your help. I appreciate it.

31) Page 13, line 392: typo ‘phosphorous’.

Response: The suggested correction has been made. Thank you so much for your help. I appreciate it.

32) scheme 27: compound 15 is actually 15a as R is ethyl in this reaction.

Response: The suggested correction has been made. Thank you so much for your help. I appreciate it.

33) Scheme 28: compound 5f should be 5e.

Response: The suggested correction has been made. Thank you so much for your help. I appreciate it.

34) Page 15, line 443: Compound 9b should be 9i.

Response: The suggested correction has been made. Thank you so much for your help. I appreciate it.

35) Schemes 30 and 31: the captions to these schemes call for a ‘proposed mechanism’, but in reality the sketch does not individuate any mechanism. The same point holds also in other parts of this manuscript where a mechanism picture is mentioned,. In contrast, the discussion about the mechanism is generally only of marginal significance. The same also in scheme  37.

Response: The suggested correction has been made. Thank you so much for your help. I appreciate it.

36) Scheme 31: Scheme 31 does not reflect the text that is mentioning the formation of 'a complex product mixture'. However, compounds 7 and 3 are not included in the reaction scheme and species 8 is written among the reagent species.

Response: The suggested correction has been made. Thank you so much for your help. I appreciate it.

37) Scheme 34: The conditions for the decomposition of 15a, as well as those used to attempt the decomposition of the related species 15b, should be specified in the Scheme.

Response: The suggested correction has been made. Thank you so much for your help. I appreciate it.

A comparison of schemes 34 and 35 shows that the compound numbered as 15b is used to individuate two different species.

Response: The suggested correction has been made. Thank you so much for your help. I appreciate it.

38) Page 18, line 531: the sentence ‘a small amount’ is vague; the quantity of  3f,g should be made explicited.

Response: The suggested correction has been made. Thank you so much for your help. I appreciate it.

39) Page 18, line 544: The term periododiphosphane is a scarcely used and outdated name to identify P2I4. The correct name is hypodiphosphorous tetraiodide or, perhaps even better, diphosphorus tetraiodide.

Response: The suggested correction has been made. Thank you so much for your help. I appreciate it.

40) Page 18, line 550: typo, tertraphenyldiphosphine dioxide.

Response: The suggested correction has been made. Thank you so much for your help. I appreciate it.

41) Page 18, line 556: ‘reaction of molecule 4c with lithium in THF at 150 °C’. Please double check the reaction temperature) perhaps 50 °C?).

Dear Reviewer, because the procedure in reference was not straightforward, I deleted this information and this scheme and added another ref, which dealt with attractive reducing agent SmI2.

42) Page 19, line 564: The two organic compounds may be hardly considered as suitable catalysts for this reaction. The effective catalyst should be indicated.

Response: The suggested correction has been made, I hope. Thank you so much for your help. I appreciate it.

43) Page 19, lines 577 and 578: The specifications ‘where R = Aryl’ and ‘where R = Alkyl’ is redundant and may be easily omitted.

Response: The suggested correction has been made. Thank you so much for your help. I appreciate it.

44) Page 19, line 580: wrong abbreviations ‘AlkLi, ArylLi’.

Response: The suggested correction has been made, I hope. Thank you so much for your help. I appreciate it.

45) Page 23, line 698: ‘using copper as catalyst’, is incorrect as the effective catalyst of the reaction is copper(I) chloride. Details about the catalyzed reaction should be provided.

Response: The suggested correction has been made, I hope. Thank you so much for your help. I appreciate it.

46) The numbering scheme adopted by the authors does not help the readability of the manuscript. The compounds are generally classified by their P-P or P-O-P architectures and then individuated by adding a letter (or sometime an index!?) to the number. This fact is in principle correct but, when the class of compounds is expanded to include all the letters of the alphabet, as it happens for the hypodiphosphoric acid esters (3), the chosen system becomes very heavy and hard to be followed by the reader. This reviewer would then suggest to reconsider the adopted numbering system. Likely, the use of some extended tables listing the compounds sharing a similar structural motif could be helpful for the reader.

Response: The suggested correction has been made, I hope. I have added figure 1. Thank you so much for your help. I appreciate it.

47) In general, as many abbreviations have been used in the manuscript, a clear definition of each of them should be entered after the first use of such abbreviation. A table summarizing ALL the used abbreviations is missing but it is highly necessary and must be included in the new version of this manuscript.

Response: The suggested correction has been made, I hope. I have added “Abbreviations” in the end of manuscript. Thank you so much for your help. I appreciate it.

48) The conclusion section is too short and unsatisfactory and does not indicate which could have been the real motivation inspiring this work. The two sentences forming this section do not add anything to the significance of this compendium and do not offer any justification for this review. The section is therefore limited to a pair of tautological sentences which stress the same identical concept without adding any contribution to the discussion. As a whole, this review consists of a long collection of reactions but does not offer to the reader any critical view of the studied subject matter.

Response: The suggested correction has been made, I hope. Thank you so much for your help. I appreciate it.

49) In references 53, 80, 92 the title of the journal and the volume should be italicized. Refererences 80 and 96 are incomplete as they lack of the title of the cited article.

Response: The suggested correction has been made, I hope. Thank you so much for your help. I appreciate it.

Some improvements in the writing have been made. I have revised the whole manuscript carefully and tried to avoid any grammar or syntax errors. Besides, I have asked several skilled authors of English language papers to check the English. Thank you so much for your help. I appreciate it

Yours sincerely, Jacek Nycz (The enclosed pdf covers some pictures showing answers from the articles)

Reviewer 2 Report

The review manuscript entitled: “The Synthesis of Hypodiphosphoric Acid and Derivatives With P-P Bond, Including Esters and Diphosphine Dioxides”

by Dr. Jacek E. Nycz

reports an interesting an overview on the literature regarding hypodiphosphoric acid, also known as hypophosphoric acid, and its derivatives from the beginning of studies on this compound to date.

The manuscript describes extensively what is known and the topic makes it worthy of publication.

However, a careful revision is needs, because along the manuscript there are many typos that make make reading difficult.

For example:

  • at rows 219 and 226 is written diethyl phosphate (5a) and phosphate anion (5a’) instead of phosphite (or phosphonate);
  • at row 302: Scheme 53 instead of Scheme 17
  • at row 318: Scheme 45 instead of Scheme 9
  • at row 362: molecule 4b instead of 11b
  • at row 431: 5e instead of 5f
  • at row 443: 9b instead of 9i
  • Scheme 431: insert formulas 3 and 7

And so on.

For these reasons the manuscript must be carefully checked prior to publication.

Author Response

A point-by-point response to comments from reviewers and editor

Reviewer #2

The review manuscript entitled: “The Synthesis of Hypodiphosphoric Acid and Derivatives With P-P Bond, Including Esters and Diphosphine Dioxides”

by Dr. Jacek E. Nycz

reports an interesting an overview on the literature regarding hypodiphosphoric acid, also known as hypophosphoric acid, and its derivatives from the beginning of studies on this compound to date.

The manuscript describes extensively what is known and the topic makes it worthy of publication.

Response: We thank the reviewer for the positive comments, constructive feedback, and recommendation for our work.

However, a careful revision is needs, because along the manuscript there are many typos that make make reading difficult.

Response: Some improvements in the writing have been made. I have revised the whole manuscript carefully and tried to avoid any grammar or syntax errors. Besides, I have asked several skilled authors of English language papers to check the English. Thank you so much for your help. I appreciate it.

For example:

  • at rows 219 and 226 is written diethyl phosphate (5a) and phosphate anion (5a’) instead of phosphite (or phosphonate);
  • Response: The suggested correction has been made. Thank you so much for your help. I appreciate it.
  • at row 302: Scheme 53 instead of Scheme 17
  • Response: The suggested correction has been made. Thank you so much for your help. I appreciate it.
  • at row 318: Scheme 45 instead of Scheme 9
  • Response: The suggested correction has been made. Thank you so much for your help. I appreciate it.
  • at row 362: molecule 4b instead of 11b
  • Response: The suggested correction has been made. Thank you so much for your help. I appreciate it.
  • at row 431: 5e instead of 5f
  • Response: The suggested correction has been made. Thank you so much for your help. I appreciate it.
  • at row 443: 9b instead of 9i
  • Response: The suggested correction has been made. Thank you so much for your help. I appreciate it.
  • Scheme 431: insert formulas 3 and 7
  • Response: The suggested correction has been made. Thank you so much for your help. I appreciate it.

And so on.

For these reasons the manuscript must be carefully checked prior to publication

Some improvements in the writing have been made. I have revised the whole manuscript carefully and tried to avoid any grammar or syntax errors. Besides, I have asked several skilled authors of English language papers to check the English. Thank you so much for your help. I appreciate it

Yours sincerely, Jacek Nycz

Reviewer 3 Report

The review entitled " The Synthesis of Hypodiphosphoric Acid and Derivatives With P-P Bond, Including Esters and Diphosphine Dioxides " presents various ways of synthesizing the titled organophosphoric acids and their derivatives containing the P-P bond.

The author covers  in this paper   a number of approaches  related to the syntheses of hypodiphosphoric acid, followed by the synthesis of hypodiphosphoric acid and its derivatives. The syntheses of diphosphine dioxides are  also described.

This is an interesting topic , worth presentation as an review compilation. The author clearly shows how effective is the synthesis of the mentioned above derivatives.  I am fully confident that this manuscript can be published as is if you will find if the English language and style are acceptable. Even though in my rating I indicated that I did not feel qualified to judge the English language and style of the manuscript, I gave 4 stars for using correct and legible English because I had not problems to follow the manuscript content in its present form.  At the same time , however,  I would like to propose the  introduction  to the manuscript  of short comments on stereochemical aspects in the chemistry of these derivatives (see for example Schemes 18, 26 ,28). Therefore I  suggest to accept this paper  with  the suggested minor additions.

Author Response

A point-by-point response to comments from reviewers and editor

Reviewer #3

The review entitled " The Synthesis of Hypodiphosphoric Acid and Derivatives With P-P Bond, Including Esters and Diphosphine Dioxides " presents various ways of synthesizing the titled organophosphoric acids and their derivatives containing the P-P bond.

The author covers  in this paper   a number of approaches  related to the syntheses of hypodiphosphoric acid, followed by the synthesis of hypodiphosphoric acid and its derivatives. The syntheses of diphosphine dioxides are  also described.

This is an interesting topic , worth presentation as an review compilation. The author clearly shows how effective is the synthesis of the mentioned above derivatives.  I am fully confident that this manuscript can be published as is if you will find if the English language and style are acceptable. Even though in my rating I indicated that I did not feel qualified to judge the English language and style of the manuscript, I gave 4 stars for using correct and legible English because I had not problems to follow the manuscript content in its present form. 

Response: I thank the reviewer for the positive comments, constructive feedback, and recommendation for our work.

At the same time , however,  I would like to propose the  introduction  to the manuscript  of short comments on stereochemical aspects in the chemistry of these derivatives (see for example Schemes 18, 26 ,28). Therefore I  suggest to accept this paper  with  the suggested minor additions.

Response: The suggested correction has been made. A small paragraph has been added to Introduction. Thank you so much for your help. I appreciate it.

Some improvements in the writing have been made. I have revised the whole manuscript carefully and tried to avoid any grammar or syntax errors. Besides, I have asked several skilled authors of English language papers to check the English. Thank you so much for your help. I appreciate it

Yours sincerely, Jacek Nycz

Reviewer 4 Report

The review of Nycz is on the synthesis of hypodiphosphoric acid and derivatives.

The review is too wordy and it could be better structured.

English usage could also be improved significantly. It is recommended to rephrase some parts by a native speaker.

Part 2: Hypodiphosphoric Acid:

This part is too detailed. In my opinion, data on X-ray structures and Raman signs, etc. should not be given, only the conclusions should be highlighted as this is not an original article but a review. 

This part needs to be shortened.

Part 3: Hypodiphosphoric Acid Esters:

This part is also too long and could be improved by restructuring in a more logical way. Also, there some inconsistencies and mistakes, some (but not all) examples are listed here:

  • in the structures of the compounds "R" should mean alkyl or aryl groups instead of alkoxy and aryloxy groups, as the structures are misleading in their current form.
  • Scheme 3: M=?
  • The style of the Schemes are very diverse, the should be unified (throughout the review!)
  • Some Schemes could be combined, such as Schemes 6 and 7.
  • Scheme 8: compounds that are referred to should not be omitted from the scheme.
  • Scheme 10 doesn't fit logically to its current place
  • Line 238: ratio 9:1 in the text is 5:1 on the scheme
  • Scheme 12: the reactions of compounds and 5 are presented several times - these should be grouped or combined when possible. For eg. Schemes 12 and 13 could also be combined showing 2 different reaction paths.
  • Scheme 14: why compound 7 is omitted?
  • Scheme 15 is again on the reaction of 4 and 5.
  • Line 302 refers to Scheme 53, but only 46 schemes are presented in this manuscript.
  • Line 318 also refers to a scheme that is not in this article.
  • etc.

The same problems apply to parts 4-8:

  • a logical order should be followed when presenting the different methods
  • a unified style should be followed 
  • some parts are too detailed, some other are omitted which makes the review more difficult to follow.

Conclusion part is too short, containing no additional information as compared to the Abstract. This part should be more informative.

In summary, the review needs serious editing of structure and a moderate editing of English usage. It may be reconsidered after major revision.

Author Response

A point-by-point response to comments from reviewers and editor

Reviewer #4

The review of Nycz is on the synthesis of hypodiphosphoric acid and derivatives.

The review is too wordy and it could be better structured.

Response: I thank the reviewer for the positive comments, constructive feedback, and recommendation for our work.

English usage could also be improved significantly. It is recommended to rephrase some parts by a native speaker.

Response: Some improvements in the writing have been made. I have revised the whole manuscript carefully and tried to avoid any grammar or syntax errors. Besides, I have asked several skilled authors of English language papers to check the English. Thank you so much for your help. I appreciate it

Part 2: Hypodiphosphoric Acid:

This part is too detailed. In my opinion, data on X-ray structures and Raman signs, etc. should not be given, only the conclusions should be highlighted as this is not an original article but a review. 

This part needs to be shortened.

  • Response: The suggested correction has been made. Thank you so much for your help. I appreciate it.

Part 3: Hypodiphosphoric Acid Esters:

This part is also too long and could be improved by restructuring in a more logical way. Also, there some inconsistencies and mistakes, some (but not all) examples are listed here:

  • Response: The suggested correction has been made. Thank you so much for your help. I appreciate it.
  • in the structures of the compounds "R" should mean alkyl or aryl groups instead of alkoxy and aryloxy groups, as the structures are misleading in their current form.
  • Scheme 3: M=?
  • Response: The suggested correction has been made. Thank you so much for your help. I appreciate it.
  • The style of the Schemes are very diverse, the should be unified (throughout the review!)
  • Response: The suggested correction has been made. Thank you so much for your help. I appreciate it.
  • Some Schemes could be combined, such as Schemes 6 and 7.
  • Response: The suggested correction has been made. Thank you so much for your help. I appreciate it.
  • Scheme 8: compounds that are referred to should not be omitted from the scheme.
  • Response: The suggested correction has been made. Thank you so much for your help. I appreciate it.
  • Scheme 10 doesn't fit logically to its current place
  • Response: The suggested correction has been made. Thank you so much for your help. I appreciate it.
  • Line 238: ratio 9:1 in the text is 5:1 on the scheme
  • Response: The suggested correction has been made. Thank you so much for your help. I appreciate it.
  • Scheme 12: the reactions of compounds and 5are presented several times - these should be grouped or combined when possible. For eg. Schemes 12 and 13 could also be combined showing 2 different reaction paths.
  • Response: The suggested correction has been made. Thank you so much for your help. I appreciate it.
  • Scheme 14: why compound 7is omitted?
  • Response: The suggested correction has been made. Thank you so much for your help. I appreciate it.
  • Scheme 15 is again on the reaction of 4and 5.
  • Response: The suggested correction has been made. Thank you so much for your help. I appreciate it.
  • Line 302 refers to Scheme 53, but only 46 schemes are presented in this manuscript.
  • Response: The suggested correction has been made. Thank you so much for your help. I appreciate it.
  • Line 318 also refers to a scheme that is not in this article.
  • Response: The suggested correction has been made. Thank you so much for your help. I appreciate it.

The same problems apply to parts 4-8:

  • Response: The suggested correction has been made. Thank you so much for your help. I appreciate it.
  • a logical order should be followed when presenting the different methods
  • Response: The suggested correction has been made, I hope. Thank you so much for your help. I appreciate it.
  • a unified style should be followed 
  • Response: The suggested correction has been made. Thank you so much for your help. I appreciate it.
  • some parts are too detailed, some other are omitted which makes the review more difficult to follow.

Conclusion part is too short, containing no additional information as compared to the Abstract. This part should be more informative.

  • Response: The suggested correction has been made. Thank you so much for your help. I appreciate it.

In summary, the review needs serious editing of structure and a moderate editing of English usage. It may be reconsidered after major revision.

Some improvements in the writing have been made. I have revised the whole manuscript carefully and tried to avoid any grammar or syntax errors. Besides, I have asked several skilled authors of English language papers to check the English. Thank you so much for your help. I appreciate it

Yours sincerely, Jacek Nycz

Reviewer 5 Report

This review is devoted to the synthesis of hypodiphosphoric acid and derivatives with P-P bond. Although these compounds are of interest as important natural compounds, the data presented in this review shows that this topic has not been very popular in recent years: only 32 papers have been published this century and not a single one in the last two years; the majority papers were published in 20th century. The results of a number of old works presented in the review raise doubts about their reliability, since the final structures have not been confirmed by modern spectral methods, for example, line 214 “The products were isolated by vacuum distillation and identified by vapor phase chromatography”; line 221 “He found the formation of high-boiling by products, which, based on the refractive index values”; line 254. “The product was … finally characterized by Raman spectroscopy”. There are a lot of factual errors in the text, when the yields on the Schemes (for example, in Schemes 9, 11 and others) do not coincide with the data described in the text of the paper. The Schemes are drawn in different formats, they need to be unified. A number of schemes (eg 4, 7, 15, 17 etc) are unsuccessful; the yields of the final compounds are not given, and it is difficult to understand which of the reaction directions is the main one, and which structures were simply marked in the text of the papers. The last names of the authors on lines 482-490 are incorrect. The title of the papers in References (58-66, 97) should be given in English; they can be found in Chemical Abstracts references.

The paper can be recommended for publication in the journal ‘Molecules’ after major revision.

Author Response

A point-by-point response to comments from reviewers and editor

Reviewer #5

This review is devoted to the synthesis of hypodiphosphoric acid and derivatives with P-P bond. Although these compounds are of interest as important natural compounds, the data presented in this review shows that this topic has not been very popular in recent years: only 32 papers have been published this century and not a single one in the last two years; the majority papers were published in 20th century. The results of a number of old works presented in the review raise doubts about their reliability, since the final structures have not been confirmed by modern spectral methods, for example, line 214 “The products were isolated by vacuum distillation and identified by vapor phase chromatography”; line 221 “He found the formation of high-boiling by products, which, based on the refractive index values”; line 254. “The product was … finally characterized by Raman spectroscopy”.

There are a lot of factual errors in the text, when the yields on the Schemes (for example, in Schemes 9, 11 and others) do not coincide with the data described in the text of the paper. The Schemes are drawn in different formats, they need to be unified. A number of schemes (eg 4, 7, 15, 17 etc) are unsuccessful; the yields of the final compounds are not given, and it is difficult to understand which of the reaction directions is the main one, and which structures were simply marked in the text of the papers.

Response: I thank the reviewer for the positive comments, constructive feedback, and recommendation for our work.

The last names of the authors on lines 482-490 are incorrect.

Dear Reviewer, unfortunately, I do not understand what mistake I made. In the article I quoted, the name is written like this. To be sure, I've pasted the header of this article below:

The title of the papers in References (58-66, 97) should be given in English; they can be found in Chemical Abstracts references.

Response: I thank the reviewer for the positive comments, constructive feedback, and recommendation for our work.

The paper can be recommended for publication in the journal ‘Molecules’ after major revision.

Some improvements in the writing have been made. I have revised the whole manuscript carefully and tried to avoid any grammar or syntax errors. Besides, I have asked several skilled authors of English language papers to check the English. Thank you so much for your help. I appreciate it.

Yours sincerely, Jacek Nycz

Round 2

Reviewer 4 Report

The ms of Nycz got somewhat better after the first round of revision.

However, I have the feeling that too many suggestions remained unanswered. 

English usage is still poor, just a few examples:

  • "importance to understanding the number of.."
  • "attracted some academic interests"
  • "during working on condensing agents"
  • "some critical discussion"
  • "an ambident nucleophiles"

The review still needs some restructuring and rephrasing.

- Lines 45-52 is a new part inserted, but it is unnecessary.

- The introduction claims that the review is highlighting mechanistic considerations, however it is not really the case. Although some schemes are cited as "proposed mechanism", the scheme shows possible reaction routes not mechanisms.

- page 3 still contains too much irrelevant data

- in the structures "R" still mean R and OR. 

- Schemes can still be combines (eg. Schemes 13 and 14) 

- Scheme 15: 7a is still missing

etc.

In conclusion, the requested major revision was not fulfilled. 

I recommend publication only after a careful and thorough revision.

Author Response

A point-by-point response to comments from reviewers and editor

Reviewer #4

The ms of Nycz got somewhat better after the first round of revision.

Response: I thank the reviewer for the positive comments, constructive feedback, and recommendation for my work.

However, I have the feeling that too many suggestions remained unanswered. 

English usage is still poor, just a few examples:

"importance to understanding the number of.."

Response: The suggested correction has been made. New, more understandable sentences I added. Thank you so much for your help. I appreciate it.

"attracted some academic interests"

Response: This fragment has been deleted. Thank you so much for your help. I appreciate it.

"during working on condensing agents"

Response: This fragment has been deleted. Thank you so much for your help. I appreciate it.

"some critical discussion"

Response: This fragment has been deleted. Thank you so much for your help. I appreciate it.

"an ambident nucleophiles"

Response: The suggested correction has been made. Thank you so much for your help. I appreciate it.

The review still needs some restructuring and rephrasing.

- Lines 45-52 is a new part inserted, but it is unnecessary.

Dear Reviewer. The new part I added was due to the request of Reviewer #3. “I would like to propose the  introduction  to the manuscript  of short comments on stereochemical aspects in the chemistry of these derivatives (see for example Schemes 18, 26 ,28).”

„Among the compounds presented in this review (Fig. 1) some are diastereoisomers, the existence of which is sometimes overlooked. International Union of Pure and Applied Chemistry in Compendium of Chemical Terminology Gold Book Version 2.3.3 from 2014 defines diastereoisomers as “two non-superposable configurational units that correspond to the same constitutional unit are considered to be diastereomeric if they are not mirror images.” They are characterized by their differences in physical properties, including different spin systems in NMR spectra, which can be used to differentiate the diastereomers from each other.”

Response: The suggested correction has been made. Thank you so much for your help; however, now I would like to warmly apologize to Reviewer #3 because I deleted this part and only left one sentence. Due to both Reviewers, I added a new, more understandable sentence “The NMR spectroscopy can also be used to differentiate the diastereoisomers from each other if they are present.”

The introduction claims that the review is highlighting mechanistic considerations, however, it is not really the case. Although some schemes are cited as "proposed mechanism", the scheme shows possible reaction routes, not mechanisms.

Response: The suggested correction has been made. Thank you so much for your help. I appreciate it.

page 3 still contains too much irrelevant data

Response: The suggested correction has been made. Thank you so much for your help. I appreciate it.

in the structures "R" still mean R and OR. 

Response: The suggested correction has been made. Please see Fig. 1 and the whole manuscript. I agree now the manuscript is more understandable. Thank you so much for your help. I appreciate it.

Schemes can still be combines (eg. Schemes 13 and 14) 

Response: The suggested correction has been made. Mentioned Schemes 13 and 14 are combined. Thank you so much for your help. I appreciate it.

Scheme 15: 7a is still missing

Response: The suggested correction has been made. The 2,2'-oxybis(5,5-dimethyl-1,3,2-dioxaphosphinane 2-oxide) anhydride 7a (now 10a) presented on Scheme 14 (present version) was isolated in the case when equimolar proportions of used reagents 7b’ and 5a, were used. However, when the Authors used 50% of the excess of nucleophile 7b the anhydride 7a (now 10a) was not isolated. Thank you so much for your help. I appreciate it.

Some improvements in the writing have been made. I have revised the whole manuscript carefully and tried to avoid any grammar or syntax errors. Besides, I have asked several skilled authors of English language papers to check the English. Thank you so much for your help. I appreciate it.

Yours sincerely, Jacek Nycz

Reviewer 5 Report

Dear Editor,

The authors have corrected almost all the shortcomings noted in the review. It remains to correct the errors of yields in Schemes 9 and 10, which do not coincide with the data given in the text of the paper. 

Author Response

A point-by-point response to comments from reviewers and editor

Reviewer #5

Response: I thank the Reviewer for the positive comments, constructive feedback, and recommendation for my work.

The authors have corrected almost all the shortcomings noted in the review. It remains to correct the errors of yields in Schemes 9 and 10, which do not coincide with the data given in the text of the paper.

Response: The suggested correction has been made. Thank you so much for your help. I appreciate it.

Re: Scheme 9

Dear Reviewer, the mentioned in the manuscript Authors curried out two reactions between 3-methyl-5-(trichloromethyl)-1,2,4-oxadiazole and diethyl phosphonate (7a) in the presence of triethylamine in a diethyl ether environment (first) and in ethyl alcohol (second) [47]. In the first reaction (in a diethyl ether environment), nucleophile 7a attacked the halogen of the trichloromethyl group of the 3-methyl-5-(trichloromethyl)-1,2,4-oxadiazole yielding dehalogenated 5-(dichloromethyl)-3-methyl-1,2,4-oxadiazole with 67% yield (Scheme 9). In the second experiment (in ethyl alcohol), the Authors isolated 5-(dichloromethyl)-3-methyl-1,2,4-oxadiazole with a 91% yield. Possibly because the in situ generated electrophile 5b reacted faster with ethanol than with nucleophile 7a. As a result, the nucleophile 7a in higher concentration undergo more efficiently a reaction with 3-methyl-5-(trichloromethyl)-1,2,4-oxadiazole.

The suggested correction has been made. Thank you so much for your help. I appreciate it.

Some improvements in the writing have been made. I have revised the whole manuscript carefully and tried to avoid any grammar or syntax errors. Besides, I have asked several skilled authors of English language papers to check the English. Thank you so much for your help. I appreciate it.

Yours sincerely, Jacek Nycz
